# Wild *Lactobacillus casei* Group Strains: Potentiality to Ferment Plant Derived Juices

**DOI:** 10.3390/foods9030314

**Published:** 2020-03-09

**Authors:** Elena Bancalari, Vincenzo Castellone, Benedetta Bottari, Monica Gatti

**Affiliations:** Department of Food and Drug, University of Parma, Viale delle Scienze, 49/A, 43124 Parma, Italy; elena.bancalari@unipr.it (E.B.); vincenzo.castellone@unipr.it (V.C.); monica.gatti@unipr.it (M.G.)

**Keywords:** wild *Lactobacillus casei*-group strains, plant derived juices fermentation, impedometric analysis, exopolysaccharides, probiotic

## Abstract

Plant derived beverages have recently gained consumers’ interest, particularly due to their intrinsic functional properties. They can also act as non-dairy carriers for probiotics and prebiotics, meeting the needs of lactose allergic/intolerant people and vegans. Direct fermentation of fruit and vegetables juices by probiotic lactic acid bacteria could be a tool to increase safety, shelf-life, nutrients bioavailability and to improve sensorial features of plant derived juices. This study aims to screen wild *Lactobacillus casei*-group strains isolated from dairy matrices for probiotic features, such as acid and bile salts resistance, and test them for the potentiality to ferment celery and orange juices. Strains’ ability to produce exopolysaccharides (EPS) in situ is also checked. These evaluations were performed for the first time in fruit and vegetables matrices by means of an impedometric analysis, recently shown to be a suitable and rapid method to measure microorganisms’ growth, acidification performances and EPS production. This study allowed the selection of three potentially probiotic *L. casei*-group wild strains able to ferment fruit and vegetable juices and also producing EPS. These strains with three-in-one abilities could be used to produce new functional fermented plant derived juices.

## 1. Introduction

Demand for plant derived products has increased in recent years due to their recognized health benefits [1,2,3]. Although recommendations suggest the consumption of fresh fruit over fruit juices and derivatives, sometimes supplying the markets with fresh fruit can be tricky because of the high intrinsic perishability of fruit. In this optic, non-sweetened fruit and vegetable juices can be valid alternatives to whole fruit equivalents in meeting dietary requirements, improving cardiovascular health and lowering incidence of several chronic non-communicable diseases [4,5,6]. For this reason, the attention of industry has focused on producing juices or formulated beverages with nutritional properties, such as richness in bioactive compounds and nutrient factors [7]. However, industrial production of these kinds of product requires particular attention. In fact, to obtain safe juices with a prolonged shelf life, treatments are needed to stabilize them. Even though mild technologies are sometimes applied to these products [8,9,10], most frequently thermal treatments are used [11,12]. These could modify juices’ nutritional properties, by degrading micronutrients and lowering contents of vitamin C, provitamin A and other nutritional factors like antioxidants and phytochemicals [13]. A useful alternative, able to maintain and/or improve the safety, nutritional, sensory and shelf life properties of fruits and vegetables, is lactic acid fermentation [14,15,16,17,18]. Traditionally, most fermented products were based on milk, but fermentation of non-dairy matrices is gaining increasing attention [19]. Thanks to the ability of lactic acid bacteria (LAB) to ferment also plant derived and mildly acid substrates such as fruit and vegetable juices [18,20]. Dairy fermentation is usually driven by starter LAB species such as *Streptococcus thermophilus, Leuconostoc mesenteroides, Lactococcus lactis* and *Lactobacillus delbrueckii* spp. *bulgaricus* [21,22]. However, species belonging to the *Lactobacillus casei*-group are frequently used as adjunctive and/or secondary starters to improve fermented products’ characteristics. Species belonging to this group are also well known for their probiotic traits [21,23]. Probiotics are defined as “live microorganisms which when administered in adequate amounts confer a health benefit to the host” [24]. Considering that plant derived juices have proved to be promising carriers for probiotics [25], direct fermentation of vegetables and fruit juices by probiotic LAB could be a tool to increase safety, shelf-life, nutrient bioavailability and to improve sensorial features of plant derived juices [19]. Furthermore, consumers’ demand for non-dairy probiotic foods is constantly increasing due to drawbacks related to dairy foods such as allergy, lactose intolerance and cholesterol content, as well as a revolution in living standards, eating habits (i.e., vegetarians and strict vegans), religious beliefs and augmented health awareness [21,25]. Finally, from a technological point of view, plant derived probiotic products could offer a direct prebiotic activity and may help to deliver the probiotic organisms to the target sites [21]. Prebiotics have been defined as a non-digestible food ingredient that beneficially affect the host by selectively stimulating the growth and/or activity of one or a limited number of bacteria in the colon, and thus improve host health [26]. Among prebiotics, many exopolysaccharides (EPSs) from LAB have been already studied for their potential prebiotic activities [27,28]. EPS production seems to help the survival of probiotic bacteria during the gastrointestinal transit [29], suggesting EPS production as an interesting property to be considered for the selection of putative probiotic strains. The aim of this study is to evaluate the ability of wild dairy *L. casei*-group strains to ferment plant derived substrates and to produce EPS *in situ*, in order to select the most suitable to be used for the production of functional juices. This evaluation has been performed for the first time in these matrices by means of an impedometric analysis, recently shown to be a suitable and rapid method to measure the growth of microorganisms [30] and reveal EPS production [31].

## 2. Materials and Methods 

### 2.1. Bacterial Strains

Fifty-three wild *L. casei* group strains, previously isolated from dairy matrices and belonging to the microbial collection of the Department of Food and Drug of the University of Parma, were tested for their acid and bile salts resistance (Table 1). Ten out of the 53 analyzed strains, were tested for their ability to grow in non-dairy extracts: four *Lactobacillus casei (Lc* 2233, *Lc* 2243, *Lc* 2404, *Lc* 2410); one *Lactobacillus paracasei* (*Lp* 2306) and five *Lactobacillus rhamnosus* (*Lr* 2216, *Lr* 2299, *Lr* 2325, *Lr* 2409, *Lr* 2462). Moreover, a commercial probiotic strain, *Lactobacillus rhamnosus* GG (ATCC 53103) was used. All the strains were maintained as stock cultures at −80 °C in De Man, Rogosa and Sharpe (MRS) broth (Oxoid, Basingstoke, UK) supplemented with 20% (*v*/*v*) glycerol until use. Upon usage, bacteria were revitalized by inoculating 200 µL of thawed cultures in 6 mL of sterile MRS incubated for 24 h at 37 °C in anaerobiosis condition. Overnight cultures were counted to verify the microbial cell load (data not shown), washed with Ringer (Oxoid, Basingstoke, UK) solution and properly diluted to reach an inoculum level of 8 Log CFU/mL.

### 2.2. Determination of Acid and Bile Salts Resistance

Resistance to increasing concentration of bile salts was measured for the 53 *L. casei*-group strains (Table 1) by streaking 10 µL of overnight cultures on MRS agar, supplemented with 0.2 g/L and 0.4 g/L of bile salts and incubated at 37 °C for 48 h. Acid resistance was tested by streaking 10 µL of overnight cultures of strains on MRS agar adjusted at pH 2.5 with hydrochloride acid and incubated at 37 °C for 48 h. As a positive control, strains were grown on MRS plates. After 48 h of incubation, growth was verified by visual inspection of plates. Results were reported in Table 1 as follows: (−) absence of growth, (+) low growth, (++) abundant growth. 

### 2.3. Fruit and Vegetables Juices Extraction, pH Measurement and Microbial Enumeration

Fresh, organic and commercially matured orange, celery and red beet were purchased in a local market and quickly transported to laboratory to be processed and prepared for the experiments. Selected fruit and vegetables were cleaned and separated from peels and leaves not used to produce the extracts. Raw materials were then washed with cold water, rinsed with demineralized water and allowed to dry for 30 min at room temperature. Raw vegetables were cut in pieces and the juices extracted with a domestic juice extractor (Juice art plus 110631, RGV, Como, Italy).

pH of fresh juices was measured electrometrically, just after extraction, with a pH meter Beckman ϕ™ 300 series (Beckman Instruments, Inc. 4300 N. Harbor Blvd. Fullerton, CA 93835, USA).

Microbial counts were evaluated on fresh juices just before inoculation and after 60 h of fermentation. Ten-fold serial dilutions were prepared in 0.9% Ringer solution (VWR Chemicals, Radnor, PA, USA) and spread onto MRS agar (Oxoid, Basingstoke, UK) for LAB count, YEDC (Yeast extract, dextrose, chloramphenicol agar, Lenexa, KS, USA) for yeasts and molds, and PCA (Plate Count Agar, Oxoid, Basingstoke, UK) for total microbial count (TMC). Plates were incubated at 37 °C under aerobic condition for 24 h (72 h for YEDC). Colony forming units were finally counted and expressed as Log CFU/ mL of fresh juice.

### 2.4. Acidification Ability and EPS Production of LAB Strains in MRS and Juices

To investigate strains’ ability to grow, acidify and produce EPS, impedance measurements were performed by means of BacTrac 4300^®^ (Sylab, Generon, San Prospero, MO, Italy) in fresh juices and with MRS as a control.

The strains were 10-fold diluted in sterile Ringer solution and used to inoculate at a 2% (*v*/*v*): 18 mL of MRS and 18 mL of fresh juices. Both were equally divided into three sterilized BacTrac 4300^®^ measurement vials which were located inside the instrument and incubated at 37 °C for 60 h. 

For the evaluation of strains’ acidification ability, the M-values, which is the overall impedance variation of the media in the vials, was measured. The M-value was recorded every 10 min for 60 h and shown as M%. This value is automatically calculated by the instrument as relative change compared to a starting value. The resulting M% data were fitted to the Modified Gompertz equation to obtain the kinetic parameters Lag and yEnd, used to describe the performances of LAB both in MRS and juice. Lag is described as an adjustment period and is measured in hours. The higher the value, the bigger the time that the cells need to adapt to the growth conditions. yEnd is the highest variation of impedance recorded and it is interpreted as the maximum acidifying capacity of the strains [30].

For the evaluation of EPS production, both in MRS and juices, the E-values, which are the electrochemical double layer of the electrodes-electrolyte impedance, were measured every 10 min for 60 h. As for M%, the measured E-values are shown as E% changes compared to a starting value. As already described by Bancalari et al., the EPS production can be revealed by measuring the decrease in E% values [31]. To this end, parameter ΔE% was calculated as the difference between the maximum value reached by E% and the value recorded after 60 h of incubation. The ΔE% values were calculated from triplicate experiments.

### 2.5. Statistical Analysis

Results of impedometric measurements were statistically analyzed with a two-way ANOVA model performed using PRC GLM of SAS (SAS Inst. Inc., Cary, NC, USA), whereas SIMCA 16 (Sartorius Stedim Data Analytics, Gottinga, Germany) software was used to create a principal component analysis (PCA) biplot to get visual interpretation of the data analyzed.

## 3. Results

### 3.1. Resistance to Bile Salts and Acid Condition

Results obtained after the screening of 53 strains analyzed for their acid and bile salts resistance are reported in Table 1. Not all the tested strains resulted as resistant to both acid and bile salts. *Lc* 2337, *Lr* 1678 and *Lr* 2438 were not able to grow in any of the stress conditions, and two strains *Lc* 2406 and *Lr* 2416 could grow in MRS with HCL but not in MRS with both concentrations of bile salts. Finally, five strains (*Lc* 2407, *Lr* 2412, *Lc* 2413, *Lr* 2466, *Lp* 2092) could grow both on acid MRS and MRS with bile salts at the lowest concentration (Table 1).

Taken together, these results showed that 10 out of the 53 analyzed strains, highlighted with bold characters in Table 1, had a higher (++) resistance to all the tested stress conditions. These strains were considered for further analysis. Acid and bile salts resistance are two important traits that a microorganism should have to be considered as a probiotic able to explicate health benefits to the host gut [32,33,34]. In fact, in order to reach the gut, microorganisms must have the ability to pass through the human gastro-intestinal tract (GIT) characterized by an extremely low pH and the presence of toxic glycoconjugated bile salts [35,36]. 

It must be noted that these results can only suggest a potential probiotic activity of the chosen strains, but are not enough to define them as probiotic. 

### 3.2. Microbial Enumeration of Analyzed Juices

Extracted juices were analyzed immediately after extraction and after 60 h by plate counting. Results showed that celery and orange juices had a negligible initial TMC, 1.82 ± 0.02 Log CFU/mL, while red beet juice presented a TMC of 5.93 ± 0.03 Log CFU/mL. Due to this, red beet juice was not considered for the fermentation with LAB and further analysis. An initial high contamination level, in fact, would make it impossible to obtain a stable and safe fermented juice without a stabilization treatment (e.g., thermal treatment) able to achieve a sufficient reduction of microbial load [37,38], but also could possibly decrease potential functional properties of the juice [8,10,11]. Celery and orange juices were then inoculated with the 10 selected LAB strains and *Lr* GG, and incubated for 60 h at 37 °C. At the end of fermentation, the TMC showed a small increase. Yeasts and mold count decreased down to 0.7 Log CFU/mL in orange juice and to 1 Log CFU/mL in celery juice. 

### 3.3. Acidification Ability of LAB Strains in Fresh Juices

Impedometric analysis, performed on celery and orange juices inoculated with the ten most acid and bile salt resistant strains and *Lr.*GG, are shown in Table 2 as mean value ± SD of three replicated for each strain. 

All the tested strains were able to grow in both juices giving a measurable variation of impedometric (M%) signal.

Nevertheless, the impedometric analysis revealed a diverse growth ability of the strains in juices as compared to the one measured in MRS. These differences were evaluated by observing both the yEnd values and the pH (Table 3). 

These values, although lower than those recorded in MRS, confirmed that all the tested strains were able to duplicate, metabolize and acidify both in celery and orange juice despite the initial low pH values of 5.8 and 4 respectively. The greater decrease of pH was measured for strains growing in celery juice, resulting in higher ΔpH values (difference between initial pH and after 60 h fermentation) (Table 2). 

In particular, the strains that caused the highest decrease of pH were *Lr* 2216, *Lr*2409 and *Lr* GG. 

The low pH caused by the production of organic acids by LAB is known to act as an antimicrobial agent, making the environment not suitable for the growth of the majority of pathogenic and spoilage microorganisms [39,40]. The LAB fermentation could thus be responsible for the safety and stability of the juice, which in case of production should be further verified. 

yEnd values, which are interpreted as the maximum acidifying capacity of the strains, ranged from 4.25 to 8.66 in celery and from 4.28 to 10.22 in orange juice (Table 3). However, yEnd values are not related to ΔpH values. This is due to the fact that the impedometric technique does not depend only on pH variation but measures the complex modification of the electrical conductivity of the medium in which LAB strains develop [41]. Despite the absence of a direct correlation between yEnd and pH values, we found it interesting that *Lr* GG was the only strain showing at the same time the greatest yEnd value and the widest ΔpH. This may suggest that the metabolism of some LAB strains in fresh juices could be more complex than just lowering the pH by producing lactic acid. 

Lag values, indicating the time needed by microorganisms to adapt to the substrate, are dependent on the growth conditions and on the physiological state of the cells, therefore the results obtained are specific to the conditions tested. Lag resulted to be less than 4 h for all the strains growing in celery juice, with a low variability (ΔLag 2.18 h) among the strains. In particular, the strain with the statistically lowest Lag value (1.82 h) was *Lr* GG resulting in the fastest adaptation in celery juice (Table 3).

The strains’ behavior in orange juice was more heterogeneous, with statistically significant differences among the strains. *Lc* 2243 was the slowest adapting strain (Lag 18.59 h) followed by *Lc* 2404 (Lag 15.19 h) and *Lr* 2325 (Lag 13.31 h). The fastest adapting strain in orange juice was again *Lr* GG, with a Lag value of 6.43 h, comparable only to *Lc* 2410. The variability of Lag values among strains was statistically higher (ΔLag 12.6 h) in orange juice than in celery, suggesting the slowest adaptability of the strains in orange juice, probably due to its composition in terms of polyphenols and antioxidant, combined with the low pH [42].

However, even if at a different extent, the ability of the tested strains to grow and acidify both celery and orange juice is in agreement with Amal Bakr Shori [43], who found that several lactobacilli have a very high tolerance to a plant derived acid environment, probably because of the high nutrient content that makes them an ideal substrate for probiotic strains growth [44].

All the measured values (Lag, yEnd and pH) were then plotted together in a PCA biplot in order to get a better view of the behavior of the strains in the two plant derived juices (Figure 1). As can be seen, the strains were well separated on the first component, accordingly to the matrix.

In particular, it can be noted that the highest Lag values were found for strains fermenting orange juice, while the highest ΔpH values were recorded for the strains fermenting celery juice.

Observing the biplot it can be noted that *Lr* GG is well separated from the other strains, showing the best adapting ability (lower Lag value) and also the best acidification performances (lowest pH value and higher yEnd). 

Nevertheless, among the wild tested strains, *Lc* 2410 and *Lc* 2233 were the best performing strains in orange juice, while *Lr* 2216 was the best strain in both types of juices. 

### 3.4. EPS Production

The potentiality of the strains to produce EPS during fermentation in juices was investigated by using the capacitance value (E%) recorded by the BactTrac 4300^®^ [31]. E%, that is the double layer capacitance of the electrodes/electrolyte interface, is strongly affected by any modification of the ionic layers in the vicinity of the electrode surface [31]. For this reason, this measurement is extremely sensitive to slight alteration of the surface of the electrodes-electrode impedance, and this is the reason why it was used to detect the EPS production.

In fact, in case of EPS production, adhesion or placement in the electrodes nearby slightly alters the interface impedance by blocking the registration of electrical impedance at the area of contact, causing the decrease of the capacitance values (E%) [31]. 

EPS production was thus calculated as ΔE%, i.e., the difference between the maximum value reached by capacitance values recorded (E% max) and after 60 h of incubation. The ΔE% values were calculated from triplicate experiments. ΔE% was measured for all the strains also in MRS as a control (Table 4).

As only the strains with a ΔE% higher than 3 are considered EPS producers [31], we concluded that no strain was able to produce EPS in orange juice. In MRS 6 strains out of 11 showed a ΔE% value higher than 3 being therefore considered able to produce EPS. Four of this strains, 3 *Lb. rhamnosus* (*Lr* GG, *Lr* 2299, *Lr* 2325) and one *L. casei (Lr* 2233) were able to produce EPS also in celery juice. Interestingly, two strains, *Lr* 2409 and *Lc* 2410, were able to produce EPS in celery juice but not in MRS. The strain *Lr* 2216 that was the best performing strain in both juices was able to produce EPS in MRS but not in the juices. This would make it a good candidate for plant derived juices fermentation, but would open questions about its potential probiotic effect, as it has been demonstrated that EPS have a protective effect on probiotics transiting the GIT [45], protecting them from low pH and high bile salt concentration [28]. 

## 4. Conclusions

Our results showed that *Lr* GG, a well-known commercial probiotic strain, showed the best ability among the tested strains to ferment both orange and celery juice. On the other hand, we were able to select at least three wild strains (*Lr* 2299, *Lr* 2409 and *Lc* 2410) with good fermentation performances in both juices, also producing EPS in celery juice. These displayed characteristics open new perspectives for dairy isolates to be used as starter in plant derived juices fermentation. Furthermore, the idea of adding probiotic strains to drive the fermentation process can be a good strategy to obtain a probiotic product avoiding any thermal treatment and therefore maintaining all the intrinsic beneficial properties of the juices. In conclusion, the use of selected potentially probiotic strains with three-in-one abilities (good acidification performances, probiotic and EPS-producers) could be a valid tool to obtain a new functional fermented plant derived beverage.

## Figures and Tables

**Figure 1 foods-09-00314-f001:**
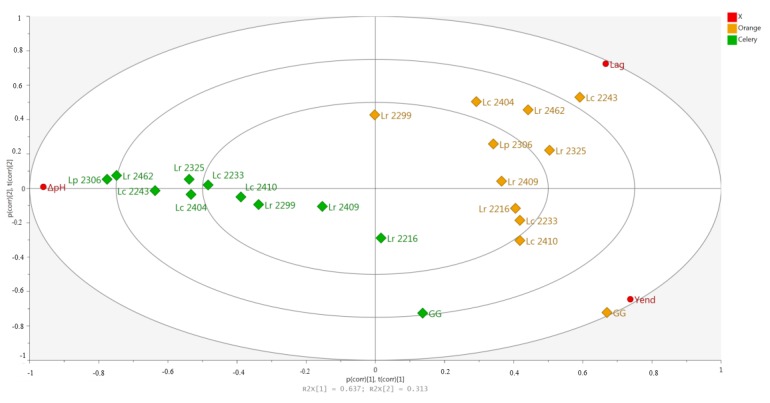
Principal component analysis (PCA) score plot (t1 vs. t2) of the first two PCs of the data set considered against the loading plot (p1 vs. p2). The variables Lag, yEnd and ΔpH are marked (red). The first component explains 64% of the variation and the second component 33%. Observations are colored according to juices.

**Table 1 foods-09-00314-t001:** Strains’ resistance to acid and bile salts. For each strain, the collection number, species and isolation matrix are given. Results are reported as absence of growth (−), low growth, (++) abundant growth. Strains written in bold were chosen for further experiments.

Strain	Species		Isolation Matrix			
				MRS HCl	MRS + bs 0.2	MRS + bs 0.4
**2233**	***L.***	***casei***	**PR cheese**	**++**	**++**	**++**
**2243**	***L.***	***casei***	**PR cheese**	**++**	**++**	**++**
2322	*L.*	*casei*	PR cheese	++	++	++
2326	*L.*	*casei*	PR cheese	++	++	++
2333	*L.*	*casei*	PR cheese	++	+	+
2337	*L.*	*casei*	PR cheese	-	-	-
**2404**	***L.***	***casei***	**PR cheese**	**++**	**++**	**++**
2405	*L.*	*casei*	PR cheese	++	++	++
2406	*L.*	*casei*	PR cheese	++	-	-
2407	*L.*	*casei*	PR cheese	++	++	-
**2410**	***L.***	***casei***	**PR cheese**	**++**	**++**	**++**
2413	*L.*	*casei*	PR cheese	++	-	+
2092	*L.*	*paracasei*	GP cheese	++	++	-
2302	*L.*	*paracasei*	PR cheese	++	++	++
2303	*L.*	*paracasei*	PR cheese	++	++	++
**2306**	***L.***	***paracasei***	**PR cheese**	**++**	**++**	**++**
2408	*L.*	*paracasei*	PR cheese	++	++	+
1019	*L.*	*rhamnosus*	PR cheese	++	++	+
1200	*L.*	*rhamnosus*	PR cheese	++	++	++
1473	*L.*	*rhamnosus*	PR cheese	++	++	+
1678	*L.*	*rhamnosus*	PR curd	-	-	-
2118	*L.*	*rhamnosus*	GP cheese	++	++	+
2190	*L.*	*rhamnosus*	GP cheese	++	++	+
2197	*L.*	*rhamnosus*	Raw milk	++	+	+
2203	*L.*	*rhamnosus*	Raw milk	++	++	++
**2216**	***L.***	***rhamnosus***	**Raw milk**	**++**	**++**	**++**
2222	*L.*	*rhamnosus*	Raw milk	+	++	+
2232	*L.*	*rhamnosus*	PR curd	++	++	+
2240	*L.*	*rhamnosus*	PR cheese	++	++	++
2246	*L.*	*rhamnosus*	PR cheese	+	++	+
2247	*L.*	*rhamnosus*	PR cheese	+	+	+
2298	*L.*	*rhamnosus*	PR cheese	++	++	++
**2299**	***L.***	***rhamnosus***	**PR cheese**	**++**	**++**	**++**
2300	*L.*	*rhamnosus*	PR cheese	++	++	++
2310	*L.*	*rhamnosus*	PR cheese	++	++	++
2323	*L.*	*rhamnosus*	PR cheese	++	++	++
**2325**	***L.***	***rhamnosus***	**PR cheese**	**++**	**++**	**++**
2334	*L.*	*rhamnosus*	PR cheese	++	++	+
2335	*L.*	*rhamnosus*	PR cheese	++	++	++
2336	*L.*	*rhamnosus*	PR cheese	++	++	++
2352	*L.*	*rhamnosus*	GP cheese	++	++	+
2362	*L.*	*rhamnosus*	GP cheese	++	++	+
2400	*L.*	*rhamnosus*	GP cheese	++	++	+
**2409**	***L.***	***rhamnosus***	**PR cheese**	**++**	**++**	**++**
2411	*L.*	*rhamnosus*	PR cheese	++	++	++
2412	*L.*	*rhamnosus*	PR cheese	++	-	+
2414	*L.*	*rhamnosus*	PR cheese	++	++	+
2415	*L.*	*rhamnosus*	PR cheese	++	++	++
2416	*L.*	*rhamnosus*	PR cheese	++	-	-
2438	*L.*	*rhamnosus*	Raw milk	-	-	-
**2462**	***L.***	***rhamnosus***	**GP cheese**	**++**	**++**	**++**
2465	*L.*	*rhamnosus*	GP cheese	++	++	+
2466	*L.*	*rhamnosus*	GP cheese	++	++	-

L.: *Lactobacillus*; PR: Parmigiano Reggiano; GP: Grana Padano.

**Table 2 foods-09-00314-t002:** Plant derived juices’ microbial cell load for yeast (YPD), lactic acid bacteria (MRS) total microbial count (TMC), and pH of fermented and unfermented juices. Counts are expressed as log CFU/mL.

T0	TMC	MRS	YPD	pH
Red beet	5.93 ± 0.03	3.22 ± 0.01	4.43 ± 0.01	6.0
Celery	1.82 ± 0.01	1.30 ± 0.03	1.18 ± 0.02	5.8
Orange	1.82 ± 0.02	1.48 ± 0.01	2.47 ± 0.01	4.0
T60	TMC	MRS	YPD	pH
Celery	4.5 ± 0.02	5.2 ± 0.1	1 ± 0.01	5.1
Orange	3.8 ± 0.1	2.6 ± 0.02	0.7 ± 0.01	3.9

**Table 3 foods-09-00314-t003:** Results of the impedometric measurements reported as mean value ± SD, for each strain in both the substrates used.

Species	Strains	Juice	Lag ± SD	yEnd ± SD	pH± SD	ΔpH
*Lb. casei*	2233	Celery	3.88 ^h^ ± 0.15	4.75 ^kijl^ ± 0.25	4.5 ^c^ ± 0	1.3
*Lb. casei*	2243	Celery	4.00 ^h^ ± 0.30	5.02 ^kijgh^ ± 0.54	4.8^a^ ± 0	1.0
*Lb. casei*	2404	Celery	3.46 ^h^ ± 0.43	4.99 ^kijh^ ± 0.12	4.7 ^b^ ± 0	1.1
*Lb. casei*	2410	Celery	3.55 ^h^ ± 0.09	5.10 ^ijgh^ ± 0.17	4.4 ^d^ ± 0	1.4
*Lb. paracasei*	2306	Celery	3.68 ^h^ ± 0.02	4.48 ^kjl^ ± 0.52	3.9 ^h^ ± 0	1.9
*Lb. rhamnosus*	2216	Celery	3.04 ^h^ ± 0.52	6.50 ^e^ ± 0.51	4.2 ^f^ ± 0	1.6
*Lb. rhamnosus*	2299	Celery	2.93 ^ih^ ± 0.44	5.16 ^igh^ ± 0.16	4 ^g^ ± 0	1.8
*Lb. rhamnosus*	2325	Celery	3.48 ^h^± 0.47	4.40 ^kl^ ± 0.53	4.4 ^d^ ± 0	1.4
*Lb. rhamnosus*	2409	Celery	3.59 ^h^ ± 0.18	5.46 ^fgh^ ± 0.47	4.3 ^e^ ± 0	1.5
*Lb. rhamnosus*	2462	Celery	3.44 ^h^ ± 0.19	4.25 ^l^ ± 0.22	4.7 ^b^ ± 0	1.1
*Lb. rhamnosus*	GG	Celery	1.82 ^i^ ± 0.08	8.66 ^b^ ± 0.03	3.9 ^h^ ± 0	1.9
*Lb. casei*	2233	Orange	7.82 ^f^ ± 0.24	7.46 ^c^ ± 0.32	3.9 ^h^ ± 0	0.1
*Lb. casei*	2243	Orange	18.59 ^a^ ± 0.42	6.43 ^e^ ± 0.51	3.8 ^i^ ± 0	0.2
*Lb. casei*	2404	Orange	15.19 ^b^ ± 0.85	5.41 ^igh^ ± 0.22	3.8 ^i^ ± 0	0.2
*Lb. casei*	2410	Orange	6.56 ^g^ ± 0.60	7.80 ^c^ ± 0.24	3.7 ^j^ ± 0	0.3
*Lb. paracasei*	2306	Orange	12.37 ^c^ ± 0.55	6.08 ^fe^ ± 0.21	3.7 ^j^ ± 0	0.3
*Lb. rhamnosus*	2216	Orange	8.40 ^ef^ ± 0.87	7.19 ^dc^ ± 0.06	3.9 ^h^ ± 0	0.1
*Lb. rhamnosus*	2299	Orange	10.43 ^d^ ± 0.40	4.28 ^l^ ± 0.26	3.7 ^j^ ± 0	0.3
*Lb. rhamnosus*	2325	Orange	13.31 ^c^ ± 1.86	6.64 ^de^ ± 0.74	3.7 ^j^ ± 0	0.3
*Lb. rhamnosus*	2409	Orange	9.53 ^ed^ ± 1.80	6.53 ^de^ ± 0.66	3.7 ^j^ ± 0	0.3
*Lb. rhamnosus*	2462	Orange	14.98 ^b^ ± 0.88	5.66 ^fg^ ± 0.55	3.7 ^j^ ± 0	0.3
*Lb. rhamnosus*	GG	Orange	6.43 ^g^ ± 0.81	10.22 ^a^ ± 0.51	3.7 ^j^ ± 0	0.3

^a–l^ Different lowercase letters by column indicate the presence of significant differences according to ANOVA (*p* < 0.001).

**Table 4 foods-09-00314-t004:** Mean value of the measured ∆E% value for each strain in MRS, Celery and Orange.

Species	Strain	∆E% MRS	∆E% Celery	∆E% Orange
*Lb. casei*	2233	3.52 ± 1.92	4.35 ± 2.20	0.11 ± 0.22
*Lb. casei*	2243	1.75 ± 0.08	0.06 ± 0.04	0.49 ± 0.83
*Lb. casei*	2404	2.47 ± 0.66	0.61 ± 0.47	0.25 ± 0.03
*Lb. casei*	2410	2.07 ± 1.13	5.56 ± 2.32	0.21 ± 0.02
*Lb. paracasei*	2306	6.25 ± 2.49	2.81 ± 0.98	0.6 ± 0.01
*Lb. rhamnosus*	2216	4.28 ± 0.99	2.51 ± 1.34	0.55 ± 0.12
*Lb. rhamnosus*	2299	6.55 ± 2.94	4.43 ± 2.21	0.81 ± 0.42
*Lb. rhamnosus*	2325	4.25 ± 1.84	6.06 ± 0.61	0.1 ± 0.03
*Lb. rhamnosus*	2409	2.37 ± 0.67	5.83 ± 2.37	0.75 ± 0.08
*Lb. rhamnosus*	2462	2.43 ± 0.51	2.88 ± 1.67	0.45 ± 0.07
*Lb. rhamnosus*	GG	4.09 ± 1.62	3.04 ± 0.86	0.67 ± 0.12

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
