# Peer review of "Wild Lactobacillus casei Group Strains: Potentiality to Ferment Plant Derived Juices"

_foods, 2020, doi:10.3390/foods9030314_

Round 1
Reviewer 1 Report
The manuscript entitled Wild Lactobacillus casei group strains: potentiality to ferment plant” ” is an interesting article.
The abstract section includes all the matters studied in the paper. The introduction justifies the research done, and the objectives have been clearly stated.
The Material and methods section has been adequately explained, but I would like to do some considerations:
1) Units of microorganisms counts: log CFU/mL-1 is not correct. It is log CFU/mL or log CFU mL-1.
2) Figure 1: This figure is entirely explained in the text. It could be deleted.
3) Table 1 is a mixture of M&M and R&D. Would it be possible to reduce its size by joining similar results?
4) Revise some decimal numbers: for example, 0,9 Ringer solution
5) Salmonella Shigella agar: Method for counting E. coli and Salmonella is not appropriate. My advice is delete all results obtained with this method.
6) Sentence and other sentences “This value is automatically calculated by the instrument as relative changes compared to a starting value. [29]. The resulting M% data were fitted to the Modified Gompertz equation to obtain the kinetic parameters Lag and yEnd, used to describe the performances of LAB 124 both in MRS and juices [29]. Lag is described as an adjustment period and is measured in hours. The highest the value, the bigger the time that the cells need to adapt to the growth conditions. yEnd is the highest variation of impedance recorded and is interpreted as the maximum acidifying capacity of the strains [29]. Only a reference should be included and correctly (not . [29].) at the end of the paragraph.
The interpretation and discussion of the results are adequate, but I would like to do some considerations:
1) Sentence: “The low pH caused by the production of organic acids by LAB is known to act as an antimicrobial agent, making the environment not suitable for the development of pathogenic and spoilage microorganisms [38,39]. So in the case of celery juice, the higher acidification would ensure the safety of the product (Table 4), while in orange juice would be ensured anyway by the very low starting pH”.This sentence is a hypothesis. Results cannot confirm that microorganisms, as for, eg. SETEC or Salmonella cannot survive in these conditions, because methods used have not been adequates.
2) Sentence: “Despite the absence of a direct correlation between yEnd and pH values we found it interesting that Lr GG showed the greatest yEnd value and the lowest ΔpH.” ΔpH of strain Lr GG was not the lowest.
“This may suggest that the metabolism of the strains in fresh juices is more complex than what can be observed by simply measuring pH decrease.” This hypothesis should be explained in more detail. Speculation!
Author Response
The manuscript entitled Wild Lactobacillus casei group strains: potentiality to ferment plant” ” is an interesting article.
The abstract section includes all the matters studied in the paper. The introduction justifies the research done, and the objectives have been clearly stated.
- The Material and methods section has been adequately explained, but I would like to do some considerations:
- Units of microorganisms counts: log CFU/mL-1 is not correct. It is log CFU/mL or log CFU mL-1.
We thank the revisor for the comment. We have corrected the units of microorganisms count in the text as suggested.
- Figure 1: This figure is entirely explained in the text. It could be deleted.
Thanks for your kind comment, the figure has been deleted.
- Table 1 is a mixture of M&M and R&D. Would it be possible to reduce its size by joining similar results?
We thank the reviewer, according to the journal’s instruction for the authors, tables should appear after the first time they are mentioned in the text. As we tried to keep the table amount as low as possible, and as we didn’t want to duplicate the table, we combined the table with the strains identification and origin with the results from resistance tests. For these reasons table 1 is presenting both M&M and R&D. In order to reduce its size, we shortened the species names and the origin by using the abbreviations explained in the table foot note: Lactobacillus is now L., Parmigiano Reggiano is now PR and Grana Padano is now GP.
4) Revise some decimal numbers: for example, 0,9 Ringer solution
We thank the reviewer for the comment and we have corrected all the decimal number in the text
5) Salmonella Shigella agar: Method for counting E. coli and Salmonella is not appropriate. My advice is delete all results obtained with this method.
We thank the reviewer, we removed methods and results concerning Salmonella and E. coli.
6) Sentence and other sentences “This value is automatically calculated by the instrument as relative changes compared to a starting value. [29]. The resulting M% data were fitted to the Modified Gompertz equation to obtain the kinetic parameters Lag and yEnd, used to describe the performances of LAB 124 both in MRS and juices [29]. Lag is described as an adjustment period and is measured in hours. The highest the value, the bigger the time that the cells need to adapt to the growth conditions. yEnd is the highest variation of impedance recorded and is interpreted as the maximum acidifying capacity of the strains [29]. Only a reference should be included and correctly (not . [29].) at the end of the paragraph.
We thank the reviewer, now only one corrected reference is included.
The interpretation and discussion of the results are adequate, but I would like to do some considerations:
1) Sentence: “The low pH caused by the production of organic acids by LAB is known to act as an antimicrobial agent, making the environment not suitable for the development of pathogenic and spoilage microorganisms [38,39]. So in the case of celery juice, the higher acidification would ensure the safety of the product (Table 4), while in orange juice would be ensured anyway by the very low starting pH”. This sentence is a hypothesis. Results cannot confirm that microorganisms, as for, eg. SETEC or Salmonella cannot survive in these conditions, because methods used have not been adequates.
We thank the reviewer for the suggestion. We have rephrased as follows: “The low pH caused by the production of organic acids by LAB is known to act as an antimicrobial agent, making the environment not suitable for the growth of the majority of pathogenic and spoilage microorganisms [39,40]. The LAB fermentation could thus be responsible for the safety and stability of the juice, which anyway in case of production should be further verified”.
2) Sentence: “Despite the absence of a direct correlation between yEnd and pH values we found it interesting that Lr GG showed the greatest yEnd value and the lowest ΔpH.” ΔpH of strain Lr GG was not the lowest.
“This may suggest that the metabolism of the strains in fresh juices is more complex than what can be observed by simply measuring pH decrease.” This hypothesis should be explained in more detail. Speculation!
We thank the reviewer for pointing out the error. We checked the values reported in table 4. The strains with the widest ΔpH were Lb. paracasei 2306 and L. rhamnosus GG. However, Lr GG showed also the highest yEnd, while Lp 2306 showed an yEnd quite low. yEnd and pH are not directly correlated (Bancalari et al. 2016) as yEnd is the highest variation of impedance recorded and is interpreted as the maximum acidifying capacity of the strains. By measuring only the pH variation, both the strains Lp2306 and LrGG would result as the best acidifying strains, but with the impedometric analysis, we were able to detect different acidifying ability between them. For this reason we also wrote the sentence that you reported as a speculation. In order to be clearer we modified the text as follows: “Despite the absence of a direct correlation between yEnd and pH values we found it interesting that Lr GG was the only strain showing at the same time the greatest yEnd value and the widest ΔpH. This may suggest that the metabolism of some LAB strains in fresh juices could be more complex than just lowering the pH by producing lactic acid”.
Reviewer 2 Report
This manuscript is a standard study on the potential of probiotic strains to survive and ferment a food matrix, in this case plant derived juices. The results will be of interest to those working in that field.
I have a few minor comments:
line 44: perhaps Lactococcus lactis and Leuconostoc mesenteroides should be mentioned as starter cultures in a dairy context.
line 54: substitute "leaving" with "living"
line 81: specify initial log CFU/ml of juice medium for each strain
line 115: change "..but and in.." with "...and with.."
line 158: In relation to this chapter, please report final log CFU/g for LAB strains in MRS and in fruit juices. Also comment on any differences between strains.
line 182 (and elsewhere): Please, define what is meant by "yEnd values"
line 193: change "For what concerns" to "Regarding"
line 201-202: change "resulted to be" to "were"
line 203: change "resulting" to "and thus"
l. 203: please, indicate that lag phase depends on specific subculturing conditions and therefore these results might not be representative undeer changed conditions.
line 209: change "a slowest" to "the slowest"
line 218: change to "a plant derived acidic environment"
line 232: change "resulted to be" to "was"
line 233: change to "..both types of juices."
line 249: change "any" tp "no"
Author Response
This manuscript is a standard study on the potential of probiotic strains to survive and ferment a food matrix, in this case plant derived juices. The results will be of interest to those working in that field.
I have a few minor comments:
line 44: perhaps Lactococcus lactis and Leuconostoc mesenteroides should be mentioned as starter cultures in a dairy context.
We thank the reviewer for the suggestion. We modified the text accordingly (Line 44-45)
line 54: substitute "leaving" with "living"
Thank you for picking the error, we corrected it.
line 81: specify initial log CFU/ml of juice medium for each strain
We thank you for the observation. All the strains were counted after the overnight incubation (data not shown) and properly diluted to reach an inoculum concentration of 108 cfu/ml for each strain.
line 115: change "..but and in.." with "...and with.."
Thank you for the suggestion, we changed the text accordingly.
line 158: In relation to this chapter, please report final log CFU/g for LAB strains in MRS and in fruit juices. Also comment on any differences between strains.
line 182 (and elsewhere): Please, define what is meant by "yEnd values"
As stated at line 124 and 192, yEnd is the highest variation of impedance recorded and it is interpreted as the maximum acidifying capacity of the strains
line 193: change "For what concerns" to "Regarding"
Thank you for the suggestion, we changed the text accordingly
line 201-202: change "resulted to be" to "were"
Thank you for the suggestion, the text was modified as follows: Lag resulted to be less than 4 hours for all the strains.
line 203: change "resulting" to "and thus"
Thank you for the suggestion, we changed the text accordingly
- 203: please, indicate that lag phase depends on specific subculturing conditions and therefore these results might not be representative undeer changed conditions.
The reviewer is right, it is well known by literature that the duration of the Lag phase depends on the strain, temperature and the substrate in which bacteria grow. As reported by Bancalari et al.,2016, the Lag phase depends on the conditions of the cultures at the moment of inoculum but also on the inoculum itself. It has been already demonstrated that the duration of Lag phase may depend on the inoculum ratio (Bancalari 2016). To be clearer we modified the text accordingly: “ Lag values, indicating the time needed by microorganisms to adapt to the substrate, is dependent to the growth conditions and on the physiological state of the cells, therefore the results obtained are specific of the conditions tested. Lag resulted to be less than 4 h for all the strains growing in celery juice, with a low variability (ΔLag 2.18 h) among the strains. In particular, the strain with the statistically lowest Lag value (1.82 h) was Lr GG resulting the fastest adapting in celery juice (Table 4).”
line 209: change "a slowest" to "the slowest"
Thank you for the suggestion, we changed the text accordingly
line 218: change to "a plant derived acidic environment"
Thank you for the suggestion, we changed the text accordingly
line 232: change "resulted to be" to "was"
Thank you for the suggestion, we changed the text accordingly
line 233: change to "..both types of juices."
Thank you for the suggestion, we changed the text accordingly
line 249: change "any" tp "no"
Thank you for the suggestion, we changed the text accordingly
Round 2
Reviewer 1 Report
The comments made in the previous review have been adequately corrected.